# How Is Spinal Cord Function Measured in Degenerative Cervical Myelopathy? A Systematic Review

**DOI:** 10.3390/jcm11051441

**Published:** 2022-03-05

**Authors:** Khadija H. Soufi, Tess M. Perez, Alexis O. Umoye, Jamie Yang, Maria Burgos, Allan R. Martin

**Affiliations:** Department of Neurological Surgery, University of California, Davis, Sacramento, CA 95817, USA; khsoufi@ucdavis.edu (K.H.S.); tmperez@ucdavis.edu (T.M.P.); aoumoye@ucdavis.edu (A.O.U.); jmmyang@ucdavis.edu (J.Y.); mdburgos@ucdavis.edu (M.B.)

**Keywords:** degenerative cervical myelopathy, cervical spondylotic myelopathy, ossified posterior longitudinal ligament, spinal cord injury

## Abstract

Degenerative cervical myelopathy (DCM) is a prevalent condition in which spinal degeneration causes cord compression and neurological dysfunction. The spinal cord is anatomically complex and operates in conjunction with the brain, the musculoskeletal system, and numerous organs to control numerous functions, including simple and coordinated movement, sensation, and autonomic functions. As a result, accurate and comprehensive measurement of spinal cord function in patients with DCM and other spinal pathologies is challenging. This project aimed to summarize the neurological, functional, and quality of life (QoL) outcome measures currently in use to quantify impairment in DCM. A systematic review of the literature was performed to identify prospective studies with at least 100 DCM subjects that utilized one or more quantitative neurological, functional, or QoL outcome measures. A total of 148 studies were identified. The most commonly used instruments were subjective functional scales including the Japanese Orthopedic Association (JOA) (71 studies), modified JOA (mJOA) (66 studies), Neck Disability Index (NDI) (54 studies), and Nurick (39 studies), in addition to the QoL measure Short-Form-36 (SF-36, 52 studies). A total of 92% (320/349) of all outcome measures were questionnaires, whereas objective physical testing of neurological function (strength, gait, balance, dexterity, or sensation) made up 8% (29/349). Studies utilized an average of 2.36 outcomes measures, while 58 studies (39%) utilized only a single outcome measure. No studies were identified that specifically assessed the dorsal column sensory pathway or respiratory, bowel, or sexual function. In the past five years, there were no significant differences in the number of total, functional, or QoL outcome measures used, but physical testing of neurological function has increased (*p* = 0.005). Prior to 2017, cervical spondylotic myelopathy (CSM) was the most frequently used term to describe the study population, whereas in the last five years, DCM has become the preferred terminology. In conclusion, clinical studies of DCM typically utilize limited data to characterize impairment, often relying on subjective, simplistic, and non-specific measures that do not reflect the complexity of the spinal cord. Although accurate measurement of impairment in DCM is challenging, it is necessary for early diagnosis, monitoring for deterioration, and quantifying recovery after therapeutic interventions. Clinical decision-making and future clinical studies in DCM should employ a combination of subjective and objective assessments to capture the multitude of spinal cord functions to improve clinical management and inform practice guidelines.

## 1. Introduction

The most common cause of spinal cord dysfunction is age-related degeneration of the discs, ligaments, and vertebrae of the cervical spine causing spinal cord compression and neurological impairment, collectively known as degenerative cervical myelopathy (DCM) [1]. The term DCM encompasses cervical spondylotic myelopathy (CSM), ossification of the posterior longitudinal ligament (OPLL), ossification of the ligamentum flavum (OLF), and degenerative disc disease [1]. Symptoms typically include numbness, paresthesias, impaired hand dexterity, weakness, unsteady gait, and sphincter dysfunction. In addition, neck pain, cervicogenic headaches, and neuropathic pain have also been associated with DCM, but the relationship of these entities with myelopathy is complex, potentially indirect, and not fully elucidated. DCM is often progressive and can manifest into severe symptoms, such as frank incontinence or quadriparesis requiring a walker or wheelchair, potentially causing affected individuals to lose their independence [1].

A number of outcome measures have been historically employed to measure the degree of neurological impairment in DCM. In 1972, Nurick proposed a popular grading system for cervical myelopathy based only on gait impairment [2]. The Japanese Orthopedic Association (JOA) score was proposed in 1985 and was later revised in several versions (most recently 1994) and has been widely adopted in Japan and East Asian countries. In 1991, Benzel et al. proposed a modified JOA (mJOA) score that replaced assessment of the “use of chopsticks” with cultural references that were more appropriate for Western countries, including “buttoning a shirt” and “eating with a spoon”. The use of the mJOA has subsequently increased, including several clinical trials and as the basis of the categorization of DCM into mild (mJOA ≥ 15), moderate (mJOA 12–14), and severe (mJOA < 12) [3]. The reliance on these outcome measures has increased to the point that clinical practice guidelines (CPGs) published by AOSpine are based on the mJOA alone, recommending surgery for moderate-severe cases and mild cases that show progressive deterioration [4]. However, reliance on the mJOA is problematic in several ways, as scores can be affected by other medical conditions, interobserver reliability is limited [5,6], and it remains unclear how to best assess patients for neurological deterioration, although comprehensive clinical assessments and quantitative microstructural MRI have been proposed [7,8].

Our current understanding of the complex anatomy and physiology of the spinal cord suggests that more accurate measurements of spinal cord function may be necessary to optimize surgical clinical decision making, the design of clinical trials, and the refinement of future CPGs. The current study aims to analyze the existing literature to determine what neurological, functional, or quality of life (QoL) outcome measures have been utilized to quantify impairment in DCM, for the purpose of identifying research trends, practice patterns, and gaps in our current knowledge.

## 2. Materials and Methods

A systematic review of the literature was performed in accordance with the Preferred Reporting Items for Systematic Reviews and Meta-Analyses (PRISMA) guidelines and the Cochrane Handbook of Systematic Reviews of Interventions [9,10], and was registered with PROSPERO (CRD42022307161). An electronic database search was performed in PubMed, Embase, and MEDLINE. Search terms were formulated with the assistance of an academic librarian using PubMed and the search strategy was adjusted for the other databases (Appendix A).

The inclusion criteria were as follows: original research studies (randomized controlled trials—RCTs, cohort studies, case series, cross-sectional and case-control studies) with at least 100 human subjects with a diagnosis of DCM, prospectively collected data (allowing for retrospective analysis of prospectively collected data), English language, and the use at least 1 quantitative/numeric outcome measure that assessed neurological, functional, or quality of life status (Table 1). DCM was defined as a degenerative pathology causing extrinsic spinal cord compression, including CSM, OPLL, OLF, and disc herniations. For the purposes of this study, functional outcome measures were defined as self-reported or administered questionnaires, scores, or ordinal scales that describe high-level functional impairments; neurological outcome measures were defined as physical testing of specific neurological functions, such as power, coordinated movements, sensation, gait, and balance; quality of life measures were defined as questionnaires that evaluated overall wellbeing. Binary measures (i.e., present/absent) were not considered quantitative, but ordinal measures with three or more levels were included. Exclusion criteria were review articles, retrospective studies, case reports, letters the editor, meta-analyses, cadaveric studies, biomechanical studies, commentaries, conference abstracts, editorials, studies with insufficient data, duplicate cohorts, and inclusion of other pathologies (tumor, inflammatory, trauma, and infection).

Three reviewers independently evaluated the search results, including performing title and abstract reviews, full-text reviews, and data extraction. Covidence (Covidence A/S, Melbourne, Australia) was used to manage citations at each step of the process. The data extracted for each study included: citation, title, year, type of study (RCT, cohort study, case series, or case-control), population studied (DCM, CSM, OPLL, disc herniation), and quantitative functional, neurological, and QoL outcome measures utilized.

Statistical analysis was performed using R v4.0.2 (R Foundation for Statistical Computing, Vienna, Austria). Data were tabulated and summary statistics were calculated for each category of outcome measures. To assess for recent trends, *post-hoc* analyses arbitrarily compared studies published in the past five years (2017 or later) and those published before 2017 using Student’s *t* tests for numeric variables and *z* tests for proportions. Results were considered statistically significant for *p* < 0.05 due to the exploratory nature of this study.

## 3. Results

The electronic database search yielded a total of 1958 unique citations, of which 362 studies were retained after title and abstract review (Figure 1). After full-text review, 148 studies met the eligibility criteria and were included in this study (Appendix A). Of all included studies, there were 84 (57%) cohort studies, 39 (26%) case series, 14 (10%) RCTs, and 11 (7%) cross-sectional studies (Figure 2). Sixty-six (45%) of the studies were multicenter or involved shared databases. The majority of studies (52%) utilized CSM as the preferred terminology to refer to their study population, whereas 44 (30%) used DCM, and 13 (9%) utilized OPLL, cervical spondylosis, or cervical compression myelopathy (Figure 2).

A total of 349 outcome measures were utilized by the 148 included studies, with an average of 2.36 outcome measures per study. Fifty-eight studies (39%) utilized only a single outcome measure, 28 (19%) used 2 measures, 27 (18%) used 3 measures, 25 (17%) used 4 measures, while 10 (6.8%) utilized 5 or more outcome measures (Figure 3). Of all outcome measures, 92% (320/349) were questionnaires or ordinal scales (functional or QoL), whereas objective physical testing of neurological function (strength, gait, balance, dexterity, or sensation) made up 8% (29/349).

Functional outcome measures were the most common type of instrument employed, with all studies employing at least one of these measures and an average of 1.68 measures per study. The most frequently used functional assessments were JOA (71), mJOA (66), the Neck Disability Index (NDI) (54), and the Nurick grade (39) (Figure 4, Table 2). A total of 11 other measures were used including JOA-CMEQ, the Myelopathy Disability Index (MDI) (3), the Geriatric Locomotive Function Scale (GLFS-5), overactive bladder symptom score (OBSS), and the Cooper scale.

Neurological outcome measures were infrequently utilized, with 0.20 measured per study, while 130/148 studies (88%) did not report any objective neurological measurements. The 17 studies tested lower extremity motor function, including the 10-s step test (10SST) in 8 studies, 30m walk test (30MWT) in 4 studies, 10m walk test (10MWT) in 2 studies, and Berg Balance Scale (BBS) in 1 study (Figure 4, Table 2). Electronic gait analysis (EGA) was used by two studies and assessed parameters including stride length, velocity, stability ratio (single-stance to double stance), and variability of a self-paced walk. Upper extremity motor function was measured in 8 studies using the 10s grip and release (G&R) test and 2 studies utilized Graded Redefined Assessment of Strength, Sensibility, and Prehension (GRASSP). The assessment of sensation was only performed in 3 studies, including 2 that used GRASSP and 1 that used Pain Perception Thresholds (PPT) using the PainVision PS-2100 system.

The assessment of QoL was performed with moderate frequency, with 41% (60) of studies using such measures, with an average of 0.48 measures per study. The most used instruments were SF-36 (52), EQ-5D (9), and SF-12 (3) (Figure 4, Table 2). All of the quality-of-life measures were patient-reported surveys.

Comparing recent studies published in the last 5 years (2017–2021, inclusive, *n* = 68) with older studies (*n* = 80), the average number of neurological outcome measures used has increased from 0.06 to 0.31 (*p* = 0.005). There was no difference in the number of total (2.18 vs. 2.51, *p* = 0.1), functional (1.63 vs. 1.73 *p* = 0.55), or QoL (0.49 vs. 0.48) outcome measures employed. In the past 5 years, increased use of upper extremity motor (*p* = 0.01) and lower extremity motor testing (*p* = 0.01) was observed, whereas the use of other outcome measures was similar to earlier studies. Among studies published prior to 2017, CSM was the most common terminology used to refer to the patient population (71%), but its use has declined to 36% among studies published in the past 5 years (*p* < 0.001). In contrast, the term DCM has increased in use from 9% to 48% (*p* < 0.001). Studies focused on OPLL have also proportionally increased in the past 5 years from 1% to 15% (*p* = 0.001).

## 4. Discussion

This study provides a comprehensive review of how spinal cord function is quantified in large prospective studies of degenerative cervical myelopathy and assesses the recent trends in chosen outcome measures. The most frequently utilized instruments were functional outcome measures, which we defined as subjective scores that describe high-level impairments. Versions of the mJOA and JOA scores were used in a total of 91% (135/148) of studies, suggesting that the DCM research community has reached a consensus on the use of these measures as the primary outcomes of interest. The 1994 version of the JOA was the most popular outcome measure for East Asian populations that utilize chopsticks [11], whereas the 1991 mJOA described by Benzel was the primary measure used elsewhere [12]. The NDI, which is the cervical analogue to the widely used Oswestry Disability Index (ODI) for lower back pain, was the third most common measure utilized [13]. Interestingly, all three of these outcome measures are subjective questionnaires based on patients’ self-assessments, which are subject to response, recall, and confirmation biases. These scores are also highly affected by other disabilities (e.g., knee arthritis for gait function). Furthermore, each of these scores make an unvalidated assumption of linearity and equivalence over multiple ordinal scales (e.g., 1 point on mJOA sensation is equivalent to 1 point on gait function), and they all employ terminology such as mild, moderate, or severe impairment without objective definitions. The Nurick grade was also moderately popular and is arguably somewhat more objective, providing specific criteria for each of its six levels. However, Nurick is narrowly focused on gait and does not quantify the most common deficits experienced in DCM, namely upper extremity incoordination, weakness, and numbness. Quality of life measures were employed in 41% of studies, adding important information on the overall impact of DCM, but these were universally patient-reported questionnaires that are also highly subjective and affected by comorbidities, age, and other factors. Interestingly, objective neurological assessments based on physical testing of function were performed in only 12% of studies. In addition, approximately half of studies utilized only one to two outcome measures. In the past five years, the use of objective neurological measures has increased modestly, but otherwise no major differences were detected in the number of outcome measures used, indicating that little has changed in the design of recent studies. Overall, the body of DCM literature is largely deficient in the assessment of spinal cord function, as the vast majority of studies do not obtain any objective data, nor do they perform comprehensive measurements commensurate with the complexity of the spinal cord and the deficits caused by DCM.

The results of this systematic review are consistent with a 2013 study by Kalsi-Ryan et al., which reported a narrative literature review of outcome measures used in CSM [14]. Due to the selection of CSM as the population of interest, their results were potentially biased toward Western populations, showing Nurick (34%) and mJOA (31%) as the most commonly employed measures. In keeping with our conclusions, their study found a paucity of objective data in CSM studies and concluded by recommending one additional questionnaire (QuickDASH) and 5 objective neurological measures (Berg Balance Scale, 30MWT, GRASSP, grip dynamometer, and electronic gait analysis). Our results are also consistent with a 2016 systematic review by Davies et al. [15], which investigated the selection of post-operative outcome measures, categorized in terms of function, complications, quality of life, pain, and imaging. Their review found similar results in 108 studies with slightly different inclusion criteria (prospective studies with ≥50 subjects or retrospective with ≥200), with 90% of studies reporting functional outcomes and 29% reporting QoL measures, and a preponderance of JOA (46%) and mJOA (19%) use. Furthermore, they found only scant use of objective physical testing measures, such as grip and release (1%), 30m walking test (1%), grip strength (1%), and mean locomotion score (1%). While our review overlaps considerably with the previous reviews by Kalsi-Ryan et al. and Davies et al., approximately half of the studies we identified were published after these reviews, indicating that DCM is a highly active area of research. Furthermore, our study differs from these previous works in that we specifically sought to look at the measurement of each specific function controlled by the spinal cord to determine how well previous studies have captured this information, and to identify knowledge gaps for the design of future studies and novel measurement tools.

In stark contrast to the DCM body of literature, traumatic SCI studies have unanimously adopted the ISNCSCI (formerly ASIA) exam as the primary outcome measure [16]. This comprehensive exam of motor and sensory function has several advantages, including high reliability and objective interpretation of findings, but it is time consuming and must be performed by a trained clinician, which limits its practicality for routine office use in the more common condition of DCM. It also is not sensitive to subtle spinal cord dysfunction, including hand incoordination, gait imbalance, and bladder dysfunction. However, a broad spectrum of measures that capture all aspects of motor, sensory, and autonomic impairments have also been developed and validated for SCI patients, including SCIM, FIM, WISCI, and numerous others [17,18]. This rich foundation of outcome measures, including subjective and objective data, allows for a thorough assessment of SCI patients in research studies and clinical management. However, the deficits incurred in traumatic SCI are typically more severe than encountered in DCM, highlighting the need to develop practical and tailored assessments for DCM.

The spinal cord is anatomically complex and has a myriad of functions, including motor control (simple movements, coordination, gait, and balance), sensation (pain, temperature, light touch, pressure, vibration, and proprioception), and various autonomic functions (respiratory, bladder, bowel, and sexual function). Furthermore, emerging evidence indicates that the spinal cord itself contains much of the circuitry for these functions, containing complex neuronal networks and sensorimotor feedback loops that control coordinated movements, central pattern generators, and homeostatic mechanisms. In this context, it seems almost absurd to quantify the entirety of spinal cord function in 4 questions, but the mJOA does exactly this with ordinal scales for motor dysfunction of upper extremities, lower extremities, upper extremity sensation, and bladder dysfunction. However, this simplistic approach at least addresses the most common deficits in DCM. The 1994 version of the JOA has slightly more breadth, including questions on motor function of hands, elbows, and shoulders, and sensory function in the UE, trunk, and LE. The NDI includes two questions on pain (neck pain, headaches) and eight questions regarding various functions, such as working, driving, and reading, but these are non-specific for cervical myelopathy and easily affected by comorbidities or other impairments. QuickDASH is a questionnaire developed for upper extremity function that is potentially more suited to detect deficits in myelopathy, but also non-specific. GRASSP-Myelopathy is a shortened version of the original GRASSP assessment for SCI that is tailored to DCM, measuring sensations using monofilaments (pressure, carried via the spinothalamic pathway), hand dexterity, and upper extremity strength [19]. In the current review of DCM outcome measures, there were no instruments that measured sensory modalities, such as pin pricks, temperature, light touch, vibration, or proprioception, or respiratory, bowel, or sexual function. Ideally, DCM outcome measures could be designed that are comprehensive, sensitive to mild pathology, responsive to changes, specific to myelopathy (rather than other neurological or physical deficits), valid, reliable, and objective. However, the design of such instruments is extremely challenging and needs to strike a balance between being comprehensive and practical. Given the current lack of such measures, we strongly endorse the use of multiple subjective and objective measures for future studies, clinical management, and CPGs, such as JOA or mJOA, NDI, QuickDASH, SF-36 or EQ-5D, grip strength, hand dexterity, multi-modal sensory testing, hand intrinsic power, gait, and balance testing.

This study was subject to several limitations. This review focused only on quantitative functional, neurological, and QoL outcome measures, omitting pain, range of motion, imaging, electrophysiological, and non-quantitative (binary or qualitative) outcomes that are potentially of interest. This limitation was intentional to focus this review narrowly on spinal cord function, while the excluded outcomes have numerous complexities that would benefit from their own detailed exploration. For example, pain may occur as a result of myelopathy, but also has many other potential sources, such as the nerve roots (radiculopathy), joints and ligaments (arthropathy), vertebrae (spondylosis), and muscles (myopathy and spasm). The exclusion of binary variables was necessary as a vast number of studies reported the presence or absence of symptoms and signs, which would have required careful full-text reviews to identify and led the inclusion of many additional studies. However, such binary variables are routinely used in clinical decision making (e.g., the presence of hyperreflexia, dysdiadochokinesia, or gait ataxia) and may constitute useful measurements. We also excluded retrospective and smaller prospective studies due to resource limitations and the length of the manuscript, as these studies often suffer from less thoughtful design, but we may have missed important contributions and additional outcome measures. Another limitation of this study is the English language requirement, which possibly excluded international studies, potentially biasing the results and missing useful outcomes. In addition, we created definitions for “neurological” and “functional” outcome measures to differentiate these terms, but these have variable and overlapping use in the literature. Finally, we describe physical testing of neurological function as “objective”, but these tests involve varying degrees of subjectivity (e.g., grading power from 0 to 5) and are also indirect, or surrogate, measures of spinal cord function as they also depend on the brain, peripheral nerves, and musculoskeletal systems to perform physical tasks. Electrophysiology of the spinal cord arguably offers the most “objective” measures, but has only modest sensitivity for myelopathy, cannot test complex functions (e.g., hand dexterity), and is not widely used in practice [20].

In summary, this systematic review of DCM outcome measures revealed that the majority of large prospective studies utilize a small number of outcome measures. Measurements typically include functional and QoL questionnaires, but frequently lack any objective confirmation of neurological impairment, limiting their accuracy and comprehensiveness in measuring spinal cord function. Novel outcome measures should be developed and validated that incorporate subjective and objective information and encompass the numerous functions of the spinal cord, weighting them according to their importance to patients. However, spinal cord function is difficult to measure, as it depends intrinsically on the brain, peripheral nervous system, and other body systems for its input and output, and there is a lack of ground truth information to compare novel outcome measures against. A concerted effort is needed to augment existing methods and develop new tools for quantifying disease in DCM, for the purpose of improving diagnosis, measuring severity, and monitoring patients for deterioration. Such an effort will facilitate improved clinical decision-making and standardization of practice, in addition to improving the robustness and validity of clinical research studies.

## Figures and Tables

**Figure 1 jcm-11-01441-f001:**
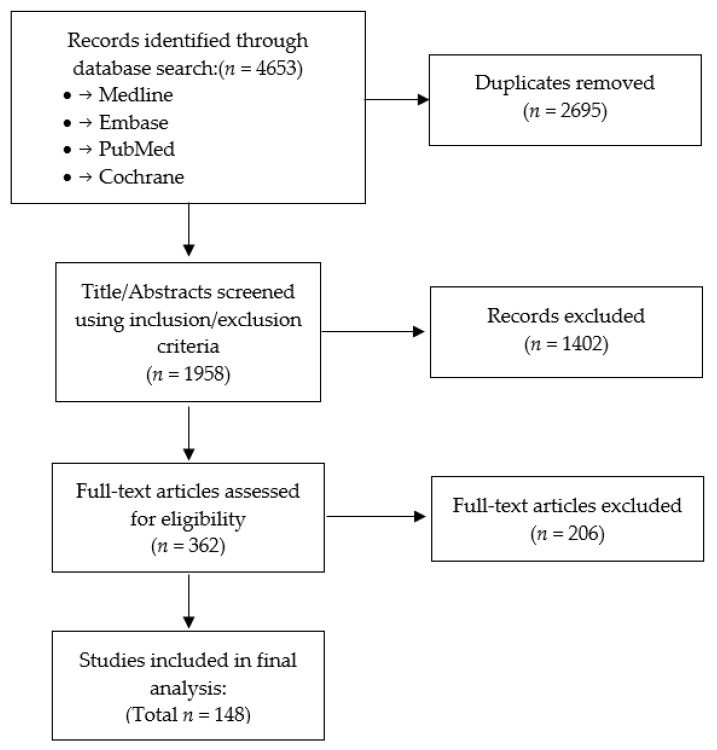
PRISMA flow diagram of systematic review. Abbreviations: PRISMA: Preferred Reporting Items for Systematic Reviews and Meta-Analyses.

**Figure 2 jcm-11-01441-f002:**
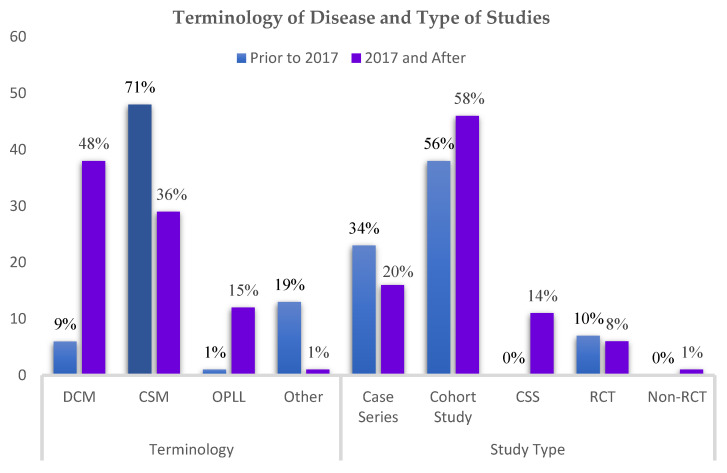
Terminology of Disease and Type of Study prior to 2017 (*n* = 68) compared to 2017 and after (*n* = 80). Abbreviations: DCM: Degenerative Cervical Myelopathy; CSM: Cervical Spondylotic Myelopathy; OPLL: Ossification of Posterior Longitudinal Ligament; CSS: Cross-Sectional Study; RCT: Randomized Controlled Trial.

**Figure 3 jcm-11-01441-f003:**
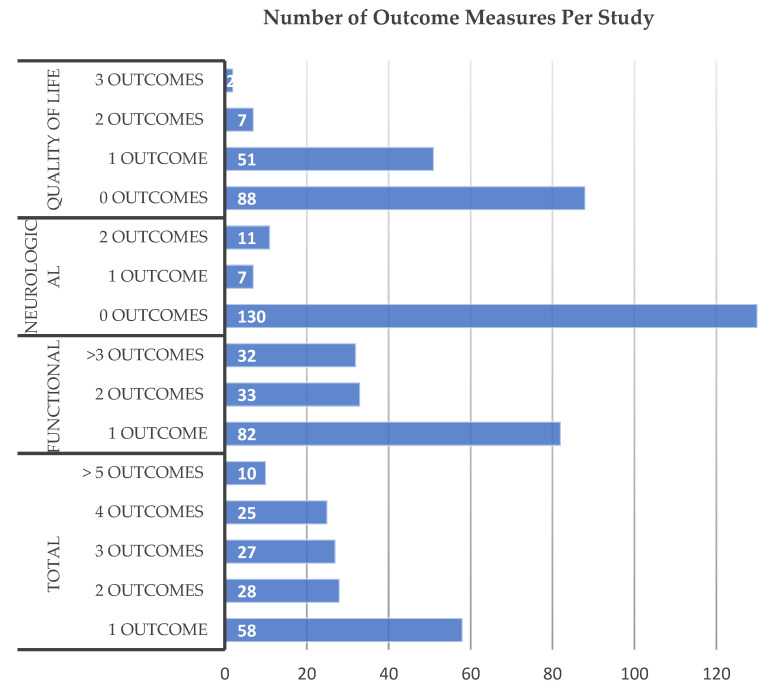
The number of total, functional, neurological, and quality of life outcome measures utilized per study in the identified literature.

**Figure 4 jcm-11-01441-f004:**
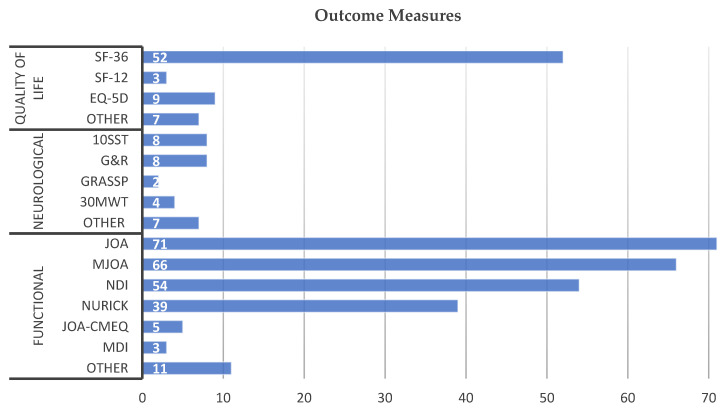
Outcome measures classified into functional, neurological, and quality of life. Abbreviations: SF-36/SF-12: Short Form 36 or 12; EQ-5D: EuroQol-5 Dimension Survey; 10SST: 10 s step test; G&R: 10 s Grip and Release; GRASSP: Graded Redefined Assessment of Strength, Sensibility, and Prehension; 30MWT: 30-m walk test; NDI: Neck Disability Index; MDI: Myelopathy Disability Index; JOA-CMEQ: Japanese Orthopedic Association Cervical Myelopathy Evaluation Questionnaire.

**Table 1 jcm-11-01441-t001:** Summary of design elements of the systematic review, in population, intervention, comparison, outcomes, and study design (PICOS) format.

PICOS Element	Criteria Used in Systematic Review
Population	Studies that analyzed patients with DCM, defined as degenerative pathology causing extrinsic spinal cord compression, including CSM, OPLL, OLF, and disc herniations. Studies were excluded if they included patients with other pathologies, including neoplastic, infectious, inflammatory, and trauma, or if they included patients without signs of myelopathy (e.g., only neck pain or radiculopathy)
Intervention	No specific intervention was required for inclusion in this review.
Comparison	No specific comparison was required for inclusion in this review.
Outcomes	Functional outcome measures, defined as self-reported or administered questionnaires, scores, or ordinal scales that describe high-level impairments. Neurological outcome measures, defined as physical tests of specific neurological functions. Quality of Life outcome measures, defined as overall measures of wellness Excluded outcome measures: pain, range-of-motion, radiographic, electrophysiologic, and non-quantitative measures
Study Design	Prospective collection of data ≥100 patients with a diagnosis of DCM Original research studies including RCTs, cohort studies, case series, and case-control studies English language Measured at least 1 quantitative outcome measure

Abbreviations: CSM: cervical spondylotic myelopathy; DCM: degenerative cervical myelopathy; OLF: ossified ligamentum flavum; OPLL: ossified posterior longitudinal ligament; PICOS: population, intervention, comparison, outcomes, study design; RCT: randomized controlled trial.

**Table 2 jcm-11-01441-t002:** Subjective and objective measurements of specific spinal cord functions or pathways in the existing literature. The number of studies that used each measure are in parentheses. Subjective assessments included functional or QoL questionnaires or ordinal scales, whereas objective assessments were defined as physical measurements of specific neurological functions.

Category of Spinal Cord Function	Specific Function or Pathway	Subjective Assessments (e.g., Questionnaires)	Objective Assessments (e.g., Physical examination)
Motor	Non-specific	SF-36 (52) NDI (54) EQ-5D (9) JOA-CMEQ (5) MDI (3) SF-12 (3) GLFS-5 (1) FIM (1)	<none>
	Strength	SF-36 (52) JOA-CMEQ (5) MDI (3) EMS (1) Ranawat (1)	GRASSP (2) MRC (1) Berg Balance (1) Grip dynamometer (1) ISNCSCI LEMS (1)
	Hand dexterity	JOA (71) mJOA (66) JOA-CMEQ (5) EMS (1)	G&R (8) GRASSP (2)
	Gait	JOA (71) mJOA (66) SF-36 (52) Nurick (39) EQ-5D (9) JOA-CMEQ (5) MDI (3) SF-12 (3) Cooper (1) GLFS-5 (1) EMS (1) Ranawat (1)	10SST (8) 30MWT (4) 10MWT (1) 10MRT (1) EGA (1)
	Balance	<none>	10SST (8) BBS (1)
Sensory	Non-specific	JOA (71) mJOA (66) JOA-CMEQ (5)	<none>
	Dorsal columns (light touch, vibration, proprioception)	<none>	<none>
	Spinothalamic (pin prick, temperature, pressure)	<none>	GRASSP (2) PPT (1)
Autonomic	Bladder	JOA (71) mJOA (66) JOA-CMEQ (5) OBSS (1)	<none>
	Bowel	<none>	<none>
	Respiratory	<none>	<none>
	Sexual	<none>	<none>

Abbreviations: FIM: Functional Independence Measure; MRC: Medical Research Council; EMS: European Myelopathy Score; GLFS-5: Geriatric Locomotive Function Scale; 10MWT: 10m Walk Test; 30MWT: 30m Walk Test; 10 MRT: 10m Run Test; 10SST: 10s step test), EGA: Electronic Gait Analysis; PPT: Pain Perception Testing; OBSS: Overactive Bladder Symptom Score; BBS: Berg Balance Scale; ISNCSCI LEMS: International Standards for Neurological Classification of Spinal Cord Injury Lower Extremity Motor Score.

## Data Availability

All data supporting reported results can be found within the manuscript.

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
