# Peer review of "How Is Spinal Cord Function Measured in Degenerative Cervical Myelopathy? A Systematic Review"

_jcm, 2022, doi:10.3390/jcm11051441_

Round 1

Reviewer 1 Report

This study summarizes the trends in the use and frequency of functional outcome measures for the spinal cord. It also mentions the strengths and problems of each component. The most frequent component was the JOA score, and neurological assessment was found to be less frequent.

This review will be helpful in organizing and using outcome assessment of spinal cord function in DCM. 

The review is generally well-written, organized, easy to read  and may be useful for reconsidering outcome assessment in the future.

Author Response

Reviewer 1:

This study summarizes the trends in the use and frequency of functional outcome measures for the spinal cord. It also mentions the strengths and problems of each component. The most frequent component was the JOA score, and neurological assessment was found to be less frequent.

This review will be helpful in organizing and using outcome assessment of spinal cord function in DCM.

The review is generally well-written, organized, easy to read  and may be useful for reconsidering outcome assessment in the future.

  • Thank you for the positive comments.

Reviewer 2 Report

The review on DCM by Soufi et al. nicely summarizes the most often used outcome measures in DCM and discusses potential limitations and future needs for the field.

While the review is well-written and gives a nice overview on DCM outcome measures, I have several critical issues which should be addressed. In general, the use of the word "objective" and "measures of spinal cord function" are misleading throughout the whole manuscript.

  • None of the listed tests are actually purely objective. Physical assessments of neurological function are often biased by the testers (e.g. for strength) or the patient themselves (e.g. for sensation). I'd suggest to omit this throughout the whole manuscript. True objective measures of spinal cord function comprise electrophysiological measures of different fiber tracts, i.e. CST, STT and DC. Unfortunately, the review was intentionally not focusing on these true objective measures as stated in the limitation section.
  • In addition, "measures of spinal cord function" would in my mind also directly relate to electrophysiological measures of the function of particular spinal pathways. The measures the authors used for spinal cord function would just indirectly relate to spinal cord function, but more directly to actual motor, sensory or ANS function. I'd suggest to rephrase where necessary.
  • Several used references in the Discussion sections are wrong and need to be double-checked. See my specific comments below.
  • Can the authors please better elude/conclude on what this review actually adds on top of the already two reviews on DCM outcomes, i.e. Davies et al., 2016 and Kalsi-Ryan et al.,2013?

Abstract

  • JOA, mJOA, NDI abbreviations in abstract not introduced
  • What's the difference between functional and physical assessments?
  • Sensitive outcome measures are not only important to monitor therapeutic but also surgical intervention success.

Intro

  • Please elude better on the novelty of this review in comparison to Davies et al., 2016 and Kalsi-Ryan et al., 2013. I'd suggest to also do that in the Abstract.

Methods

  • Line 95 replace "power" with "strength"
  • The classification into functional, neurological and some instances even physical assessment is hard to follow and not clear. E.g. testing of sensation is not an objective physical test since it's based on subjective perception of the patient.
  • The subgrouping into functional and neurological seems arbitrary as there is a huge overlap. Maybe divide into questionnaire-based function and physical examination of function?!?
  • Table 1: neck pain was introduced as a primary symptom of DCM, but listed in the Table as non-myelopathic symptom. Please clarify.

Results

  • 7% of the studies were cross-sectional ones, but those studies were not introduced as inclusion criteria. E.g. a cohort study (57%) can be a cross-sectional study. Please clarify.
  • Terminology percentages don't add up to 100%.
  • Close to half of the full-text articles were not eligible. I assume this was also based on the exclusion criteria. Or what's the difference between the inclusion and eligibility criteria? This is unclear.
  • Why was the cut-off for terminology comparison chosen to be 2017? What was the actual time period of the search? I'd expect the sample of the studies prior to 2017 be much bigger than the newer ones > 2017. Can you please also provide the amount of studies for the two time periods?
  • Line 169: were the functional measures counted to the questionnaire outcome measures? Again, this subgrouping of outcome measures is confusing. Were the functional ones only questionnaire-based? Why did the authors decide to divide into functional and neurological? Neurological function belongs to functional. E.g. gait, sensation etc. would definitely belong to functional assessments. Please clarify or even change subgrouping according to a meaningful and straight forward terminology.
  • Line 211: "Prior to 2017, CSM was the most common "terminology" for the study population (71%),…"
  • Table 2: the GRASSP (monofilament testing) and the PPT are subjective on not objective measures since they depend on the participants' rating of distinct sensations. Objective spinothalamic tract testing (STT) would include electrophysiological measures such as pain-related evoked potentials. In addition, I'd argue that the monofilament testing of the GRASSP assesses predominantly dorsal column and not STT function (see also line 314). Only PPT assesses STT function.

Discussion

  • Line249: please omit the word "objective" when talking about neurological examination. These physical assessments are often biased by the testers (e.g. for strength) or the patient themselves (e.g. for sensation). I'd suggest to omit this throughout the whole manuscript. True objective measures of spinal cord function comprise of electrophysiological measures of different fiber tracts, i.e. CST, STT and DC. Unfortunately, the review was intentionally not focusing on these true objective measures as stated in the limitation section.
  • Line 251: what is exactly meant by assessments of spinal cord function? Please provide the particular assessment.
  • The ASIA examination performed in traumatic SCI is also not an objective tool. Again, heavily depending on patient report (light touch, pinprick) or clinicians for muscle strength testing.
  • Autonomic function in SCI can be captured by the ISAFSCI (Wecht et al., 2021 - PMID: 34108833). Please include this study in line 289. In addition, it seems that Nurick 1972 is not the right reference for this statement?!?
  • In addition, Ref 19 also doesn't seem to be correct for the statements of spinal circuitries. Either the authors cite a textbook reference here or nothing at all, since it should be common knowledge.
  • Ref 20 as a reference for the GRASSP-Myelopathy is also wrong. -> The one from Suk would be more appropriate: Kalsi-Ryan et al., 2020 PMID: 31792501
  • Line 321: I'm somewhat disappointed to read that the authors endorse the use of multiple tests which are not addressing any sensory function.
  • Can the authors potentially explain why terminology was switched from CSM to DCM?

Author Response

Reviewer 2:

The review on DCM by Soufi et al. nicely summarizes the most often used outcome measures in DCM and discusses potential limitations and future needs for the field.

While the review is well-written and gives a nice overview on DCM outcome measures, I have several critical issues which should be addressed.

  • Thank you for the positive comments. We have tried to adequately address the issues you have raised.

 In general, the use of the word "objective" and "measures of spinal cord function" are misleading throughout the whole manuscript.

  • While we understand the concern raised by the reviewer, we feel the use of the word objective is reasonable given our clear definition in the methods. In medicine, the word “objective” is widely used to describe the physical examination and ancillary tests, whereas purely subjective information is that which is gained through the patient’s history (e.g. the SOAP note: https://www.wolterskluwer.com/en/expert-insights/what-are-soap-notes). The neurological exam is considered “objective” data, although we acknowledge the reviewer’s concern that it is not strictly objective. In response, we have carefully selected our language throughout the manuscript to state that physical testing of neurological function is objective in comparison with questionnaires and self-reported scores, and expanded the discussion of this issue to completely address the reviewer’s concern.

None of the listed tests are actually purely objective. Physical assessments of neurological function are often biased by the testers (e.g. for strength) or the patient themselves (e.g. for sensation).

  • We agree with the reviewer that there is a subjective component to many of the tests, but measures such as a GRIP-dynamometer, 10-second grip and release, 10-meter walk test, etc. are quite objective. In general, EP methods also have a mild degree of subjectivity (e.g. identifying the N20 peak in SSEPs, which can be challenging in myelopathy) and indirectness, depending on the brain and peripheral nerves. The physical measures we identified through this systematic review are far more objective than the questionnaires that make up the vast majority of outcome measures used in the literature and current clinical practice guidelines, which is an astonishing finding that has important implications. This is akin to a surgeon recommending surgical treatment for a patient with DCM without performing a physical examination to confirm the presence of neurological deficits. We also note the additional limitation that physical measures of function are a surrogate for spinal cord function, as they also involve the musculoskeletal system in addition to the brain and peripheral nervous system (and metabolic, cardiovascular, other systems), and are thus also indirect but useful measures. We have tried to provide an exhaustive and balanced discussion of these issues throughout the manuscript while tailoring this paper to a clinical audience that requires a pragmatic approach.

I'd suggest to omit this throughout the whole manuscript.

  • We respectfully disagree, but as stated above we have made additional changes to the discussion that clearly indicate that these are not purely objective or direct measures of spinal cord function.

True objective measures of spinal cord function comprise electrophysiological measures of different fiber tracts, i.e. CST, STT and DC. Unfortunately, the review was intentionally not focusing on these true objective measures as stated in the limitation section. In addition, "measures of spinal cord function" would in my mind also directly relate to electrophysiological measures of the function of particular spinal pathways. The measures the authors used for spinal cord function would just indirectly relate to spinal cord function, but more directly to actual motor, sensory or ANS function. I'd suggest rephrasing where necessary.

  • While we acknowledge that EP methods directly test the integrity of white matter tracts, they have profound limitations for DCM and are not widely used in clinical practice (please refer to: https://journals.sagepub.com/doi/full/10.1177/21925682211057484 ). In a recent review of EP for DCM (URL above), the sensitivity of SSEPs and MEPs was poor to moderate to detect myelopathy, let alone quantify deficits. Furthermore, EP measures only interrogate a subset of white matter pathways in a blunt and non-specific manner that is not physiological (e.g. MEPs and SSEPs use gross stimulation of thousands of neurons simultaneously), omitting complex functions of the spinal cord gray matter and integrated feedback-loops that are involved in hand dexterity, gait, and balance. Furthermore, SSEPs are primarily carried by the dorsal columns, lacking sensitivity for ventral cord pathology specific to the STTs (for which CHEPs is superior, but not widely available). Unfortunately, the inclusion of EP measures in this review would have made this manuscript excessively long and diluted the focus from clinical measures of spinal cord function, upon which clinical studies and practice currently rely upon.

Several used references in the Discussion sections are wrong and need to be double-checked. See my specific comments below.

  • Thank you for this feedback. There was an error in the numbering that led to several errors. We have addressed these.

Can the authors please better elude/conclude on what this review actually adds on top of the already two reviews on DCM outcomes, i.e. Davies et al., 2016 and Kalsi-Ryan et al.,2013?

  • Thank you for the comment. The 2nd paragraph of the discussion already addressed this issue, but we have added additional content on how this differs from both the Kalsi-Ryan and Davies reviews. Our review was more focused on the types of outcome measures utilized, and the types of spinal cord functions that are not captured by current measurement. Furthermore, the older reviews are 6 and 9 years old, respectively, and more than half the studies that we identified were published after the Davies review.

Abstract JOA, mJOA, NDI abbreviations in abstract not introduced

  • We have added in the suggested abbreviations in the abstract.

What's the difference between functional and physical assessments?

  • We have tried to clearly define our terminology in our methods, including definitions of functional and neurological assessments. Of note, we avoided the use of the term “physical assessment” because it is ambiguous and instead used the more precise term “physical testing of neurological function”. In the literature, the terms functional and neurological assessments are not well defined and thus we selected definitions that we felt were most consistent with common use.

Sensitive outcome measures are not only important to monitor therapeutic but also surgical intervention success.

  • We agree. We did not understand if this was directed toward a specific aspect of our manuscript.

Intro

Please elude better on the novelty of this review in comparison to Davies et al., 2016 and Kalsi-Ryan et al., 2013. I'd suggest to also do that in the Abstract.

  • Please see our response above. We did not have sufficient space in the abstract to include the comparison with these older reviews.

Methods

Line 95 replace "power" with "strength"

  • We respectfully disagree with the reviewer. While the terminology of “power” vs. “strength” are often used interchangeably, according to the Medical Research Council (https://www.ukri.org/wp-content/uploads/2021/12/MRC-011221-AidsToTheExaminationOfThePeripheralNervousSystem.pdf) the term power is the preferred term used to describe the relation of a muscle group movement from a joint instead of strength, as it is as an ordinal scale rather than an absolute measurement.

The classification into functional, neurological and some instances even physical assessment is hard to follow and not clear. E.g. testing of sensation is not an objective physical test since it's based on subjective perception of the patient.

  • Thank you for your comment. Please see our responses to similar comments above. Again, we feel that physically testing a subject’s sensation (e.g. with pin prick) is far more objective than asking the patient to rate their sensory loss (e.g. mJOA sensory subscore, rated as “normal”, “mild”, “severe”, or “complete”).

The subgrouping into functional and neurological seems arbitrary as there is a huge overlap. Maybe divide into questionnaire-based function and physical examination of function?!?

  • Thank you for your comment. We searched the literature for clear definitions or differences between neurological and functional outcome measures and could not find any, in spite of the fact that both terms are widely used. As a result, we decided to create definitions of these terms that we feel are consistent with their typical use throughout the literature. We feel that we have made this clear in the methods and throughout the manuscript.

Results

Table 1: neck pain was introduced as a primary symptom of DCM, but listed in the Table as non-myelopathic symptom. Please clarify.

  • Thank you for identifying this error – we have modified table 1 to state “Studies were excluded if they included patients with other pathologies, including neoplastic, infectious, inflammatory, and trauma, or if they included patients without signs of myelopathy (e.g. only neck pain or radiculopathy)”. Neck pain was introduced as associated with DCM, but not a primary symptom.

7% of the studies were cross-sectional ones, but those studies were not introduced as inclusion criteria. E.g. a cohort study (57%) can be a cross-sectional study. Please clarify.

  • We agree and thanks for identifying this. We have updated the methods.

Terminology percentages don't add up to 100%.

  • The terminology percentages are compared prior to 2017 and 2017 and after - each category adds up to about 100 (All percentages are rounded to the nearest whole number - which would account for any small variability)

Close to half of the full-text articles were not eligible. I assume this was also based on the exclusion criteria. Or what's the difference between the inclusion and eligibility criteria? This is unclear.

  • The term “eligibility criteria” encompasses both inclusion and exclusion criteria. For many of the studies, we weren’t able to identify what outcome measures were used or if the study was retrospective from the abstract alone, requiring full-text review.

Why was the cut-off for terminology comparison chosen to be 2017? What was the actual time period of the search? I'd expect the sample of the studies prior to 2017 be much bigger than the newer ones > 2017. Can you please also provide the amount of studies for the two time periods?

  • The search did not have date limits, but we did not identify many older studies due to the other restrictions on inclusion (prospective, >100 subjects). We arbitrarily selected 5 years to reflect the trends in the most recent studies and to capture any recent changes. This was a post-hoc analysis and happened to create roughly equally groups of 68 prior to 2017 and 80 studies 2017 and after. We have added these details to the methods and results.

Line 169: were the functional measures counted to the questionnaire outcome measures?

  •  

Again, this subgrouping of outcome measures is confusing. Were the functional ones only questionnaire-based?

  • These were largely questionnaires but also included ordinal scales. We have modified the text to reflect this.

Why did the authors decide to divide into functional and neurological? Neurological function belongs to functional. E.g. gait, sensation etc. would definitely belong to functional assessments. Please clarify or even change subgrouping according to a meaningful and straight forward terminology.

  • Thank you for input and please see prior comments. We appreciate that this is somewhat controversial, but we provided clear definitions in Table 1 and the methods.

Line 211: "Prior to 2017, CSM was the most common "terminology" for the study population (71%),…"

  • Thank you for the comment, this change has been made.

Table 2: the GRASSP (monofilament testing) and the PPT are subjective on not objective measures since they depend on the participants' rating of distinct sensations. Objective spinothalamic tract testing (STT) would include electrophysiological measures such as pain-related evoked potentials. In addition, I'd argue that the monofilament testing of the GRASSP assesses predominantly dorsal column and not STT function (see also line 314). Only PPT assesses STT function.

  • Thank you for the interesting comment. The dorsal columns carry light touch, while the ventral STT carries pressure. Monofilaments stimulate both, but they are primarily a pressure stimulus, and thus we are under the impression that they are carried primarily by the STT.

Discussion

Line249: please omit the word "objective" when talking about neurological examination. These physical assessments are often biased by the testers (e.g. for strength) or the patient themselves (e.g. for sensation). I'd suggest to omit this throughout the whole manuscript.

  • Please see our comments above. We have made extensive efforts to clarify our use of the term “objective”.

True objective measures of spinal cord function comprise of electrophysiological measures of different fiber tracts, i.e. CST, STT and DC. Unfortunately, the review was intentionally not focusing on these true objective measures as stated in the limitation section.

  • Please see our comments above. We have made extensive efforts to clarify our use of the term “objective”.

Line 251: what is exactly meant by assessments of spinal cord function? Please provide the particular assessment.

  • Thank you, we have revised the sentence to read more clearly: “In addition, approximately half of studies utilized only 1 to 2 outcome measures”

The ASIA examination performed in traumatic SCI is also not an objective tool. Again, heavily depending on patient report (light touch, pinprick) or clinicians for muscle strength testing.

  • Please see our comments above. We have made extensive efforts to clarify our use of the term “objective”.

Autonomic function in SCI can be captured by the ISAFSCI (Wecht et al., 2021 - PMID: 34108833). Please include this study in line 289.

  • Thank you for the insight. We have added it as reference 18.

In addition, it seems that Nurick 1972 is not the right reference for this statement?!?

  • We have removed this from the text and replaced it with a more reflective reference.

In addition, Ref 19 also doesn't seem to be correct for the statements of spinal circuitries. Either the authors cite a textbook reference here or nothing at all, since it should be common knowledge.

  • We agree and have removed the reference.

Ref 20 as a reference for the GRASSP-Myelopathy is also wrong. -> The one from Suk would be more appropriate: Kalsi-Ryan et al., 2020 PMID: 31792501

  • Thank you, we have modified the reference.

Line 321: I'm somewhat disappointed to read that the authors endorse the use of multiple tests which are not addressing any sensory function.

  • We agree – this was an oversight on our part, and we have modified the text accordingly. We endorse that future studies attempt to acquire comprehensive assessments that include all or most of the critical functions of the spinal cord

Can the authors potentially explain why terminology was switched from CSM to DCM?

  • Yes, this was formally introduced by Nouri et al. 2017 to simplify the terminology (few people actually know what spondylosis is). We have added this to the discussion.